# Distinct Mutational Profile of Lynch Syndrome Colorectal Cancers Diagnosed under Regular Colonoscopy Surveillance

**DOI:** 10.3390/jcm10112458

**Published:** 2021-06-01

**Authors:** Aysel Ahadova, Pauline Luise Pfuderer, Maarit Ahtiainen, Alexej Ballhausen, Lena Bohaumilitzky, Svenja Kösegi, Nico Müller, Yee Lin Tang, Kosima Kosmalla, Johannes Witt, Volker Endris, Albrecht Stenzinger, Magnus von Knebel Doeberitz, Hendrik Bläker, Laura Renkonen-Sinisalo, Anna Lepistö, Jan Böhm, Jukka-Pekka Mecklin, Toni T. Seppälä, Matthias Kloor

**Affiliations:** 1Department of Applied Tumour Biology, Institute of Pathology, University Hospital Heidelberg, Cooperation Unit Applied Tumour Biology, German Cancer Research Center (DKFZ), 69120 Heidelberg, Germany; paulinepfuderer@gmail.com (P.L.P.); alexej.ballhausen@gmail.com (A.B.); l.bohaumilitzky@dkfz-heidelberg.de (L.B.); svenja.koesegi@web.de (S.K.); nico.mueller@stud.uni-heidelberg.de (N.M.); yee_lin_tang@ttsh.com.sg (Y.L.T.); kosima.kosmalla@gmx.de (K.K.); johannes.witt.96@googlemail.com (J.W.); magnus.knebel-doeberitz@med.uni-heidelberg.de (M.v.K.D.); Matthias.kloor@med.uni-heidelberg.de (M.K.); 2Department of Education and Research, Central Finland Central Hospital, 40620 Jyväskylä, Finland; maarit.ahtiainen@ksshp.fi (M.A.); jukka-pekka.mecklin@ksshp.fi (J.-P.M.); 3Department of General Pathology, Institute of Pathology, University Hospital Heidelberg, 69120 Heidelberg, Germany; volker.endris@med.uni-heidelberg.de (V.E.); albrecht.stenzinger@med.uni-heidelberg.de (A.S.); 4Institute of Pathology, University Hospital Leipzig, 04103 Leipzig, Germany; hendrik.blaeker@medizin.uni-leipzig.de; 5Department of Gastrointestinal Surgery, Helsinki University Hospital, 00290 Helsinki, Finland; laura.renkonen-sinisalo@hus.fi (L.R.-S.); anna.lepisto@hus.fi (A.L.); 6Department of Pathology, Central Finland Central Hospital, 40620 Jyväskylä, Finland; jan.bohm@ksshp.fi; 7Faculty of Sports and Health Sciences, University of Jyväskylä, 40014 Jyväskylä, Finland; 8Department of Surgical Oncology, Johns Hopkins University, Baltimore, MD 21287, USA

**Keywords:** Lynch syndrome, colorectal cancer, carcinogenesis, cancer prevention, colonoscopy screening, incident cancer, microsatellite instability, mismatch repair deficiency, mutational profiling

## Abstract

Regular colonoscopy even with short intervals does not prevent all colorectal cancers (CRC) in Lynch syndrome (LS). In the present study, we asked whether cancers detected under regular colonoscopy surveillance (incident cancers) are phenotypically different from cancers detected at first colonoscopy (prevalent cancers). We analyzed clinical, histological, immunological and mutational characteristics, including panel sequencing and high-throughput coding microsatellite (cMS) analysis, in 28 incident and 67 prevalent LS CRCs (*n* total = 95). Incident cancers presented with lower UICC and T stage compared to prevalent cancers (*p* < 0.0005). The majority of incident cancers (21/28) were detected after previous colonoscopy without any pathological findings. On the molecular level, incident cancers presented with a significantly lower *KRAS* codon 12/13 (1/23, 4.3% vs. 11/21, 52%; *p* = 0.0005) and pathogenic *TP53* mutation frequency (0/17, 0% vs. 7/21, 33.3%; *p* = 0.0108,) compared to prevalent cancers; 10/17 (58.8%) incident cancers harbored one or more truncating *APC* mutations, all showing mutational signatures of mismatch repair (MMR) deficiency. The proportion of MMR deficiency-related mutational events was significantly higher in incident compared to prevalent CRC (*p* = 0.018). In conclusion, our study identifies a set of features indicative of biological differences between incident and prevalent cancers in LS, which should further be monitored in prospective LS screening studies to guide towards optimized prevention protocols.

## 1. Introduction

Individuals with Lynch syndrome (LS), the most common hereditary colorectal cancer (CRC) syndrome, have a 50% lifetime risk of developing CRC [1]. LS is caused by pathogenic variants in one of the mismatch repair (MMR) genes *MLH1, MSH2, MSH6* or *PMS2* [2]. 

Due to loss of MMR function, base mismatches occurring during DNA replication remain uncorrected and lead to insertion/deletion mutations (indels), particularly at repetitive sequences (microsatellites). Thus, cancers arising in LS exhibit the molecular phenotype of microsatellite instability (MSI). When indel mutations hit coding microsatellites (cMS), two possible biologically relevant consequences follow: first, mutations at cMS can lead to inactivation of tumor suppressor genes, contributing to carcinogenesis [3]; second, such mutations shift the reading frame and lead to generation of frameshift peptides (FSP), rendering MSI tumors highly immunogenic [4,5,6,7,8]. 

Surveillance by colonoscopy is a recommended preventive measure in LS mutation carriers [9,10]. Colonoscopy has been shown to decrease CRC incidence and mortality [11,12,13,14]. However, in contrast to the general population [15,16,17], a substantial proportion of LS mutation carriers develop “incident carcinomas”, or “post-colonoscopy CRC” [11,18,19,20,21,22,23,24] despite regular colonoscopy. In fact, recent prospective studies [22,23,25] collecting evidence from patients under surveillance demonstrated no difference in cumulative cancer incidence up to the age of 70 years when compared to studies on retrospective cohorts without surveillance [26,27,28,29].

In parallel to technical, colonoscopy quality-related explanations for the high incidence of CRC under surveillance in LS, biological explanations have been proposed, suggesting that incident cancers may develop from a precursor lesion more difficult to detect than polypoid adenomas, such as MMR-deficient crypts [30,31,32,33]. MMR-deficient crypts are morphologically indistinguishable from normal colonic crypts, but they lack the MMR protein expression on the molecular level [33,34,35]. Like MSI CRC, MMR-deficient crypts also present with MSI and MSI-induced tumor suppressor gene mutations as a direct consequence of MSI, thus possessing the theoretical potential to develop into cancer. However, direct evidence of such a progression is not trivial to obtain, as no technical means to monitor MMR-deficient crypts exist.

In contrast to clinical characteristics [36], the molecular properties of incident cancers have not been characterized so far. We aimed to analyze the molecular characteristics of incident LS CRCs diagnosed under regular surveillance and to compare them with prevalent LS CRCs diagnosed at first colonoscopy or prior to surveillance. 

## 2. Materials and Methods

### 2.1. Patients and Tumor Samples

Carriers of pathogenic MMR variants that underwent colonoscopy surveillance with a planned 3-year interval (2 years if previous CRC) were identified from the prospectively maintained Finnish Lynch syndrome registry. Available formalin-fixed paraffin-embedded (FFPE) tumor blocks from patients who developed incident (*n* = 28) and prevalent (*n* = 7) cancers were collected from the Lynch Syndrome Biobank at the Central Finland Central Hospital, Jyväskylä, Finland. FFPE tumor tissue blocks from LS patients with prevalent CRC (*n* = 60) were collected at the Department of Applied Tumor Biology, Institute of Pathology, University Hospital Heidelberg as part of the German HNPCC Consortium. Prevalent cancers were diagnosed either at first surveillance colonoscopy or prior to surveillance initiation due to symptoms. All patients provided informed and written consent, and the study was approved by the Institutional Ethics Committee (S-583/2016). Research permission was granted by the National Authority of Health and Welfare (former TEO Dnro 1272/044/07 and Valvira Dnro 10741/06.01.03.01/2015). The DNeasy FFPE Kit was used for the isolation of tumor DNA after manual microdissection from 5–6 µm thick hematoxylin/eosin (HE)-stained FFPE tissue sections (Qiagen, Germany).

### 2.2. Histopathology Analysis

HE-stained tumor tissue sections underwent histopathological analysis by two pathologists. All tumors, from which a tumor tissue section was available, were analyzed for grade and histology features. All tumors, for which the transition from normal mucosa to invasive cancer was present on the tissue sections, were assessed for their growth pattern. In addition, presence of MMR-deficient crypts was evaluated in sections containing tumor-adjacent normal mucosa and stained for the respective MMR protein (see Section 2.4).

For the assessment of tumor growth pattern, tumors were classified in three major groups: “polypoid” (corresponding to Type Ip of the Paris classification [37]), “flat” (corresponding to IIa or IIb of the Paris classification) and “depressed” (corresponding to IIc of the Paris classification). A polypoid growth pattern was defined by a tumor having fibrovascular cores and a vertical growth that is more prominent than the transverse/horizontal growth, whereas a flat growth pattern had a more prominent transverse/horizontal growth. Both polypoid and flat growth patterns were defined as tumors that were elevated above the level of the mucosa, whereas a depressed growth pattern was defined as tumors that have the bulk of the tumor located below the level of the mucosa.

### 2.3. Mutation Analysis

Mutational analysis was performed in tumor samples, from which DNA in sufficient amount and quality could be isolated. Targeted next generation sequencing was performed as described previously on an Ion Torrent S5XL/Prime sequencer using a custom 180 amplicon panel (CRC panel) encompassing mutation HotSpot regions in 30 genes [38,39,40,41]. Data analysis was performed using the Ion Torrent Suite Software (version 5.10). Only variants with an allele frequency > 5% and minimum coverage > 100 reads were taken into account. Variant annotation was performed using Annovar (hg19 genome) [40]. Annotations included information about nucleotide and amino acid changes of RefSeq annotated genes, COSMIC and dbSNP entries as well as detection of possible splice site mutations. For data interpretation and verification, the aligned reads were visualized using the IGV browser (Broad Institute) [41]. 

MSI analysis was performed using a sensitive and specific mononucleotide marker panel (BAT25, BAT26 and CAT25) as described previously [42]. cMS mutation analysis was performed using a novel high-throughput method for quantitative fragment length analysis with 5-carboxyfluorescein-labeled primers specific for a set of 22 cMS [43] (see Appendix A for details), which were selected based on two criteria: evidence of a functional driver role of mutation [43] and potential significance as a source of immunogenic frameshift peptide neoantigens [44]. PCR products were visualized on an ABI3130xl sequencer, and the obtained results were processed using the ReFrame algorithm to obtain quantitative estimation of the frequency of the mutant alleles in tumor specimens [45]. Mutation status of *B2M* was determined by Sanger sequencing as described previously [46]. The obtained mutational data for incident cancers were compared with the mutational data for prevalent cancers published in our previous reports [32,38,45].

### 2.4. Immunohistochemical Staining and Quantification of T Cell Density

FFPE tissue sections (2–3 µm) were used for immunohistochemical staining [47,48]. Briefly, sections were deparaffinized and rehydrated and subsequently stained according to standard protocols. The following primary antibodies were used: anti-CD3 (clone PS1, dilution 1:100, Abcam, Cambridge, UK); anti-MLH1 (clone G168-15, dilution 1:300, BD Pharmingen, Heidelberg, Germany); and anti-MSH2 (clone FE11, dilution 1:100, Calbiochem, Darmstadt, Germany). As a secondary antibody, the biotinylated anti-mouse/anti-rabbit antibody (Vector Laboratories) was used at 1:100 dilution. Staining was visualized using the Vectastatin elite ABC detection system (Vector Laboratories, Burlingame, CA, USA) and 3,3′-diaminobenzidine (Dako North America Inc., Carpinteria, CA, USA) as a chromogen. For counterstaining hematoxylin was used. Stained sections were scanned using NanoZoomer S210 slide scanner (Hamamatsu, Hamamatsu, Japan) and viewed using NDP.view2 Viewing Software (Hamamatsu). Four random 0.25 mm^2^ square regions were drawn in the tumor tissue and positive cells in each region were counted using the QuPath Software, giving mean values per 0.25 mm^2^.

### 2.5. Statistical Calculations

Statistical significance of differences between binary variables was calculated using Pearson’s chi-squared test or Fisher’s exact test. Statistical significance of the association of growth pattern with time since last colonoscopy was analyzed by Kruskal–Wallis test using GraphPad Prism (V6 Version 6.07, GraphPad Software Inc., La Jolla, CA, USA.) Statistical significance of differences in mutation frequencies of cMS genes, as well as significance of differences in immune infiltration, was calculated using two-sided Wilcoxon Rank Sum test (Mann–Whitney test). Correction for multiple testing was performed using Benjamini–Hochberg procedure. *p* values smaller than 0.05 were considered statistically significant. All scripts were written in R [49], version 3.6.0, using the R Studio environment [50]. All 95% confidence intervals (CI) were calculated with modified Wald method. 

## 3. Results

### 3.1. Clinical Characteristics

Clinical data and tumor specimens (*n* = 28) were collected from 27 LS patients who developed incident CRC during the 2–3-yearly preventive colonoscopy surveillance (23 *MLH1* and 4 *MSH2* carriers, 15 females and 12 males). Sixty-seven tumors from LS patients diagnosed with CRC as prevalent cancers were used as a comparison group. Median age at diagnosis was not significantly different between patients with incident and with prevalent cancer (54.4 vs. 50.0 years, *p* > 0.05). Nineteen out of 28 (68%) incident and 33 out of 49 (67%) prevalent cancers with information on tumor localization were localized in the proximal colon. Nine out of 28 tumors arose in patients with a history of previous CRC, 3 patients had other GI cancer and 10 patients had a previous non-GI cancer (endometrium, prostate, ureter, brain). The clinical parameters of incident cancers are summarized in Table 1.

The median duration of follow-up was 8.9 years (range: 0.0–29.3 years). Twelve patients with incident cancers died during follow-up due to different reasons. Three of the 12 deceased patients died due to CRC: One patient (#11) died from a symptomatic CRC that was diagnosed only two years after previous uneventful colonoscopy. Patient #16 died from another, metachronous, CRC that was diagnosed after 6 years of not attending scheduled colonoscopies. Patient #9 developed CRC liver metastases 7 years after the operation of a T2N0 caecum cancer, though no other primary tumor was found (Table 1). 

Incident cancers presented with lower UICC stage compared to prevalent cancers (*p* = 0.0002); the majority of incident cancers were stage I, whereas the majority of prevalent cancers were stage II tumors (Figure 1A). T stage of incident cancers was significantly lower than prevalent cancers (*p* = 0.00004), and no T4 lesions were identified among incident cancers (Figure 1B). Except for one tumor identified as a polyp (Table 1, #6) and treated by polypectomy, all colonic incident tumors were identified as invasive adenocarcinomas and treated by surgery. In addition, one rectum cancer (Table 1, #10) was treated by local surgical excision.

The median time since last colonoscopy in patients under surveillance was 27 months (range: 7.3–39.5 months). Time since last colonoscopy did not correlate with the stage of tumor (Appendix A). The majority of incident CRCs developed after a colonoscopy in which no lesions were detected (21/28, 75.0%, 95% CI: 56.4–87.6%, Table 1, Figure 1C). All performed colonoscopies were successful and of high quality, with complete examination reaching the remaining colon length and bowel preparation enabling the visualization of the entire mucosal surface.

### 3.2. Histopathology of Incident Cancers

Representative HE and immunohistochemistry sections of the incident cancers were examined for microscopic pattern of tumor growth, degree of differentiation and presence of MMR-deficient crypts (Appendix A). 

Among the 22 cases assessable for the tumor growth pattern, 12 showed a flat (55.5%), 6 (27.3%) showed a polypoid and 4 (18.2%) showed a depressed growth pattern. All tumors showed at least a moderate degree of differentiation, with 16/28 (57.1%) of them exhibiting mucinous components. Among 37 prevalent cancers analyzable for growth pattern, 5 showed a flat (13.5%), 15 (40.5%) showed a polypoid and 17 (45.9%) showed a depressed growth pattern (Appendix A). The proportion of tumors with mucinous components was 35.1% (20/57) in prevalent cancers. When restricted to *MLH1* patients only, the proportion of tumors with a mucinous component was significantly higher in incident cancer (13/24, 54.16%) compared to prevalent ones (4/24, 16.6%; *p* = 0.0145). Among incident cancers, the median time since last colonoscopy differed significantly (*p* < 0.05) between tumors with different histological growth pattern (median time since last colonoscopy 36.9/27.5/19.5 months in incident cancers with polypoid/flat/depressed growth pattern, respectively; Figure 1D). Interestingly, the only polypoid tumor with a short time since last colonoscopy was preceded by an advanced adenoma at previous colonoscopy (Figure 1D).

MMR-deficient crypts were present in two of the incident cancers: in both cases, these MMR-deficient crypts were present adjacent to areas of high-grade dysplasia/carcinoma in situ (Table 1, Patient #6, Figure 2A and Table 1, Patient #22, Figure 2B). The MMR-deficient crypt in Patient #22 also showed pronounced immune infiltration (Figure 2B). 

### 3.3. Mutational Profile and MMR Deficiency Signatures in Incident Cancers

We aimed to analyze how MMR deficiency influences mutational events in incident cancers and studied MMR deficiency signatures, namely the presence of C > T transitions at CpG sites and insertion/deletion (indel) mutations in *APC* and *KRAS* mutations observed in incident cancers, and compared these to previous sequencing results obtained from prevalent CRC [32,38,51]. 

In contrast to the relatively high prevalence of *KRAS* codon 12/13 mutations among prevalent LS CRCs described previously (11/21, 52% [38]), only one codon 12 mutation was identified among the analyzed incident tumors (1/23, 4.3%; *p* = 0.0005) (Figure 3A,B, Appendix A). Moreover, no pathogenic *TP53* mutations were identified in the analyzed set (0/17), which compared to prevalent cancers (7/21, 33.3% [38]) yielded a significantly lower *TP53* mutation frequency in incident CRCs (*p* = 0.0108, Figure 3A,B, Appendix A). *KRAS* and *TP53* mutation frequencies remained significantly lower in incident compared to prevalent cancers (1/19 vs. 5/9, *p* = 0.0066 and 0/14 vs. 4/9 [38], *p* = 0.0142, respectively) also when restricting the analysis to only *MLH1* carriers from both groups. The proportion of *CTNNB1*-mutant samples (5/23, 21.7%) in incident cancers was similar to the proportion of *CTNNB1*-mutant tumors detected in prevalent cancers [51] (10/48, 20.8%; *p* = 1.0, Figure 3A,B). Here, restricting the comparison to only *MLH1* carriers yielded differing proportions (4/19, 21% in incident cancer vs. 8/16, 50% in prevalent cancer [51]), though not reaching statistical significance (*p* = 0.0896).

Ten out of 17 incident cancers presented with a total of 14 truncating *APC* mutations (Figure 3B). All 14 detected *APC* mutations represented either C > T transitions at CpG sites or insertion/deletion (indel) mutations, reflecting mutational signatures associated with MMR deficiency and arguing in favor of the early onset of MMR deficiency in LS incident CRC, prior to *APC* mutations. Importantly, the proportion of such mutations was significantly higher in incident cancers compared to prevalent cancers (100 vs. 75%, 95% CI: 74.9–100 and 58.7–86.4%, respectively; *p* = 0.0470, Figure 3C,D). 

When focusing on indel *APC* mutations alone, a significantly higher proportion of mutations was found in incident CRC compared to prevalent CRC in LS patients (64.3 vs. 16.7%, 95% CI: 38.6–83.8 and 7.5–32.3%, respectively; *p* = 0.0068, Figure 3C,E).

### 3.4. CMS Analysis in Incident Cancers

Mutation frequencies obtained from the quantitative cMS analysis were compared between incident (*n* = 28) and prevalent (*n* = 67) tumors across all genes and for each gene. Generally, the frequency of cMS mutations in all 22 analyzed genes was slightly, but significantly elevated in the group of incident cancers compared to prevalent cancers (median 0.35 in incident vs. 0.31 in prevalent tumors, *p* = 0.018, Figure 4A). Analysis restricted to only MLH1-associated cancers from both groups showed similarly high cMS mutation frequency between incident and prevalent cancers (median 0.34 in incident vs. 0.33 in prevalent tumors, *p* = 0.8721, Appendix A). As mutations at cMS presumably accumulate in association with the progression time of the tumor, we analyzed cMS mutation frequencies in association with the UICC stage. In prevalent LS CRC, we observed a significant increase in the cMS mutation frequencies from UICC I to UICC II (median for UICC I 0.28 vs. UICC II 0.36, *p* = 0.002, Figure 4B), whereas the incident LS CRC presented with high cMS mutation frequencies already in stage I tumors, and no further increase was observed in stage II tumors (Figure 4C). Importantly, the cMS mutation frequency was higher in stage I incident LS CRCs compared to stage I prevalent LS CRCs (median for UICC I in incident tumors 0.35 vs. 0.28 in prevalent tumors, *p* = 0.005, Figure 4D).

The analysis of mutations in 22 specific cMS genes revealed a significantly higher proportion of mutant alleles in two genes, *LMAN1* (0.29 vs. 0.11, *p* = 0.038) and *ELAVL3* (0.37 vs. 0.17, *p* = 0.009), and a significantly lower proportion of mutant alleles in one of the analyzed cMS located in the *RFC3* (0.03 vs. 0.19, *p* = 0.011) gene in incident cancers compared to prevalent ones (Figure 4E, Appendix A).

### 3.5. Immune Infiltration and Immune Evasion in Incident Cancers

We asked whether the early onset of MMR deficiency and the higher proportion of tumors with cMS mutations is reflected by the immune response in incident cancers, and analyzed the CD3-positive T cell infiltration in incident and prevalent LS CRC. As MMR gene-dependent differences in the immunogenicity of LS CRC have been reported before [51,52], we performed an MMR gene-wise comparison of immune infiltration, focusing on the *MLH1*-associated CRCs representing the vast majority in our incident cancer group (24/28). Dense immune infiltration was observed in both incident and prevalent tumor tissue (155 vs. 149 CD3+ cells/0.25 mm^2^, respectively) and normal mucosa, although no significant differences between incident and prevalent tumors could be detected (*p* = 0.6, Figure 5).

The pronounced immune response against MSI CRC often results in the acquisition of *B2M* mutations, the most common mechanism of immune evasion in MSI tumors leading to abrogation of HLA class I-mediated antigen presentation [48,53]. We analyzed *B2M* in incident and prevalent CRCs and found a *B2M* mutation prevalence of 20.8% (5/24) in incident CRC, which was similar to the *B2M* mutation prevalence of prevalent CRC (13/54, 24.1%; *p* = 1.0). 

## 4. Discussion

In the present study, we provide first evidence that incident CRCs in LS are distinct from prevalent cancers with regard to their clinical, histological and mutational characteristics. 

Clinically, most incident cancers were of UICC stage I/II and thus of significantly lower stage than the prevalent cancers of our control cohort. Low tumor stage, typical absence of lymph node involvement and a favorable clinical course of incident cancers observed in our study are in line with previous reports [11,18,22,36,54,55]. Only one CRC-related death was clearly associated with a primary CRC included in this study and showed signet ring cell features, associated with poor survival [56]. This mirrors the previously reported excellent survival under prospective observation [57], which could be attributed to the early detection via colonoscopy. 

Histologically, tumors with mucinous components were frequent among incident CRCs in LS (57.1 vs. 35.1% in prevalent cancers). Presentation with mucinous histology in MSI cancer has previously been associated with a high cMS mutational load [58]. The elevated cMS mutation frequency detected in incident cancers of our study (Figure 4) may therefore be responsible for a high mutational variability resulting in mixed and mucinous histology. Interestingly, tumors with non-polypoid, depressed growth pattern were not substantially enriched among incident cancers compared to prevalent ones. This, together with the rather low prevalence of *CTNNB1* mutations among *MLH1*-associated incident cancers, could indicate that the progression with immediate invasive growth was not dominant in the analyzed set of incident cancers. 

Two hypotheses might explain the observed differences: first, incident cancers may in fact predominantly develop via a distinct, MMR deficiency-initiated CRC evolution. Alternatively, incident and prevalent cancers could be two entities representing manifestations of the same evolutionary pathway, detected at different time points.

The hypothesis of incident cancers representing a distinct, MMR deficiency-initiated group of tumors is compatible with two additional observations: (1) Histologically normal MMR-deficient crypt foci were detected in the direct vicinity of two incident tumors, providing indirect evidence that MMR-deficient crypts may give rise to CRC development in LS. Clonality studies of LS tumors and adjacent MMR-deficient crypts are required to provide direct proof of such associations; (2) On the molecular level, *APC* mutations in incident CRCs showed a significantly stronger association with signatures of MMR deficiency [59] than in prevalent CRCs, indicating that MMR deficiency as an early event commonly precedes APC mutations.

Importantly, we found significantly less *KRAS* mutations in incident cancers than previous studies analyzing prevalent CRC in LS [60]. Two scenarios for the observations are possible: (1) Colonoscopy with adenoma removal may theoretically be more effective in preventing *KRAS*-mutated lesions, as *KRAS* mutations are associated with conventional adenomas [61,62]. This would imply that incident cancers may develop from other, *KRAS* wild-type lesions that are more difficult to detect. In fact, a recent study analyzing the efficacy of colonoscopy depending on the molecular subtype of tumors in the general population showed a weaker CRC risk reduction after colonoscopy for *KRAS* wild-type tumors [63]; (2) Alternatively, oncogene-activating missense mutations, which need to affect very specific nucleotides and therefore have a lower likelihood per genome replication than indel mutations, may be less frequent in tumors with rapid evolution and short progression times such as incident cancers [64]. This hypothesis could also explain the absence of *TP53* point mutations, which are generally considered late events in colorectal carcinogenesis [65], and the relative scarcity of *CTNNB1*-activating point mutations in the incident CRC of our study, which were mostly of low stage. Notably, the only codon 12 *KRAS* mutation in incident CRCs was detected in a stage I tumor.

*CTNNB1* mutation frequency in incident cancers is particularly low when restricting the comparison to *MLH1*-associated LS cancers. Somatic *CTNNB1* mutations have been previously associated with *MLH1* germline variants, and a rather low prevalence of *CTNNB1*-mutant tumors among *MLH1*-associated incident cancers compared to their reported frequency in *MLH1*-associated prevalent cancers is unexpected and could point at different routes of progression between *MLH1*-associated incident and prevalent cancers. In addition, as somatic *MLH1* and *CTNNB1* mutations seem to be non-independent events [66], large deletions of the *MLH1* gene prevailing in the Finnish population as a founder variant may have an impact on the routes of cancer progression and the likelihood of somatic *CTNNB1* mutations [67]. The mechanistic reasons behind the association between *MLH1* germline variants and *CTNNB1* somatic mutations remain to be clarified by future studies.

Conceptually, the significantly lower prevalence of *KRAS* and *TP53* mutations in incident cancers could reflect earlier sampling, assuming that these mutations represent late events in CRC evolution. Following this interpretation of incident cancers and prevalent cancers as manifestations of one and the same linear evolutionary continuum, one would expect that the amount of cMS mutations is higher in prevalent (i.e., late) than in incident (i.e., early) cancers. However, the cMS mutation load of incident CRCs in our study was always in the same range or even higher in subgroups or overall comparisons compared to prevalent CRC. Notably, significantly elevated cMS mutation frequencies were observed for incident compared to prevalent cancers of UICC stage I, although this observation needs confirmation, because the number of such cancers was small in the prevalent cancer group. This observation is not in line with the suggestion that incident and prevalent cancers represent two different time points in the same CRC progression pathway. Interestingly, despite observing clearly higher cMS mutation frequencies in stage II vs. stage I prevalent cancer, for stage III and IV no further increase in cMS mutation frequencies was observed. Limited sample size or saturation effect could be possible explanations for this observation. In addition, one may speculate that cell clones with high cMS loads and consequently also high antigen load could undergo negative selection upon tumor immunoediting, favoring outgrowth of tumor cell clones with lower cMS loads. The elimination of highly immunogenic cell clones by the immune system during MSI carcinogenesis has been previously suggested [45]. However, the link between tumor immunoediting and lower cMS mutations frequencies at higher tumor stages remains to be clarified by other studies.

We are aware that our observation of cMS mutation frequency is based on targeted analysis of a limited set of 28 common mutational targets in MSI CRC [45]. In the future, NGS-based genome-wide analyses, which have limited applicability for mutation calling of cMS, such as *TGFBR2* in individual tumors [45,68], but are superior in detecting general, genome-wide effects, are warranted to quantitatively confirm cMS mutation loads in incident CRC. 

By applying the ReFrame algorithm as a highly sensitive method to detect and quantify specific cMS mutations [45], we were able to detect significant differences in mutation frequency for three individual cMS between incident and prevalent cancers: two cMS genes, *LMAN1* and *ELAVL3*, showed significantly higher mutation frequencies in incident compared to prevalent cancers, whereas the *RFC3* cMS gene showed a significantly lower mutation frequency, thereby notably showing changes in mutation frequencies in both directions. Functionally, LMAN1 is a lectin transporting glycoproteins from the endoplasmic reticulum to the Golgi apparatus [69], RFC3 is a protein participating in DNA proliferation [70], whereas the function of ELAVL3 is less well studied. *LMAN1* cMS mutations have been previously detected in hereditary MSI adenomas and suggested to play a role in early MSI-driven carcinogenesis [69], which could explain their high frequency in incident cancers. *ELAVL3* and *RFC3* cMS genes are well conserved between humans and mice and have been shown to be mutated in extracolonic MSI tumor types [43,71]. Though the exact role of these cMS mutations in incident cancer development remains elusive and the results will require confirmation in independent tumor collections, these observations point towards biological differences between incident and prevalent cancers and thus add further support to the hypothesis of two distinct entities. 

Colonoscopy quality might be another factor responsible for the development of cancers under surveillance. In our study, colonoscopies performed prior to the examination revealing cancer with the endoscopy equipment available at the recruitment period (before the introduction of high-definition endoscopy) were documented as complete procedures fulfilling the criteria for a high-quality colonoscopy (evidence of full visualization of the remaining bowel length and adequate bowel preparation) [72]. This is in line with the previous observations by Lappalainen et al., showing no association between incident cancers and a prior colonoscopy of compromised quality [73]. Additionally, the proportion of tumors located in the proximal colon, a localization often associated with lower colonoscopy sensitivity [15,63], was identical between incident and prevalent tumors analyzed in our study, indicating that localization-related colonoscopy sensitivity alone also does not explain the occurrence of incident CRCs in LS carriers under surveillance. The adenoma detection rate (ADR) in the contributing centers for follow-up colonoscopies has also been shown to be comparable with the previous reports of recent large prospective studies [22,73]. However, as there is a certain time trend towards higher ADR after the introduction of high-definition endoscopes, it cannot be ruled out that at least a proportion of incident cancers included in this study could have been prevented if currently available technical equipment was used. It seems reasonable to assume that screening parameters such as ADR and colonoscopy interval have a significant impact on the appearance and characteristics of interval cancers. The observed correlation of the growth pattern of incident cancers with time since last colonoscopy could point at a longer dwell time of tumors with polypoid growth pattern compared to tumors with depressed growth pattern. Taking into account the limited number of samples available for this analysis, this observation needs confirmation by future larger studies, ideally also including the endoscopic images of the lesions identified during colonoscopy examinations.

In line with previous observations reported by the Prospective Lynch Syndrome Database (PLSD) [54] and other large studies [22,36], no correlation was observed between time since last colonoscopy and tumor stage among incident cancers. Previous studies reported incident cancer development in the same segment of colon, where previously a polypectomy was performed, in 20–50% of cases [36,74]. Although no information on the localization of a lesion detected at previous colonoscopy was available in this study, adenoma at previous colonoscopy was found in 25% of patients with incident cancers, which is in line with other reports [18,36,73]. 

The strength of our study is the first molecular characterization of incident cancers in LS and their comparison to prevalent CRC in LS, as well as high-resolution analysis of MMR deficiency-associated mutational events using a newly established method [45]. The weakness of the study is the analysis of incident cancers from a single country with a clinical practice of 3-year colonoscopy intervals. As no structural differences have been observed in patients from different countries in previous studies with regard to their tumor risk [51], we do not expect a major influence of the sample source on our data. Moreover, all tumors were diagnosed after an interval of 40 months or less (median 27 months), with 25% of tumors after an interval of 2 years or less from previous colonoscopy. The proportion of tumors diagnosed due to symptoms was higher among prevalent compared to incident cancer groups, and a comparison with prevalent cancers detected in patients diagnosed with LS as a result of cascade testing may deliver clearer insights into differences between incident and prevalent cancer. Another limitation of our study is that the majority of incident cancers were from patients harboring *MLH1* germline variants, and thereby represented only one of the two *MMR* genes most frequently associated with incident cancer [51]. Validation of our results in a larger international multi-center study is therefore warranted in order to include more *MSH2* pathogenic variant carriers to analyze and examine potential differences between *MLH1* and *MSH2*-associated LS, as has been suggested recently [51]. Ideally, a standardized, prospectively collected cohort of incident and prevalent CRCs allowing for a gene- and stage-matched comparison would be required to validate our results.

## 5. Conclusions

In conclusion, our study for the first time identifies a set of features indicative of biological differences between incident and prevalent cancers in LS: a lower tumor stage, a high proportion of tumors with mucinous areas, a predominance of indel mutations over point mutations and a low prevalence of *RAS* mutations. These features should further be monitored in prospective LS screening studies to guide towards optimized prevention protocols, including all available options of high-quality colonoscopy and primary prevention approaches. 

## Figures and Tables

**Figure 1 jcm-10-02458-f001:**
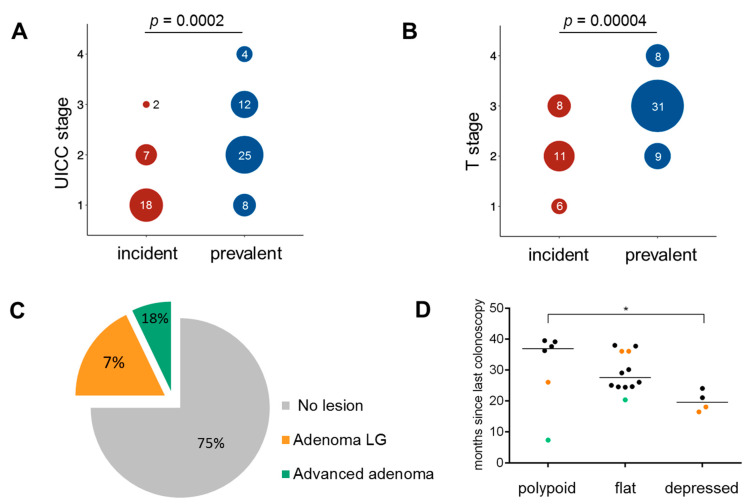
Clinical characteristics of incident cancers. (**A**,**B**) Distribution of UICC stage among incident and prevalent tumors. Tumors identified under surveillance have significantly lower UICC stage (**A**) and T stage (**B**) compared to tumors identified outside of surveillance. (**C**) Findings at previous colonoscopy in patients with incident cancers. The majority of patients with incident cancers did not present with any lesion at previous colonoscopy examination. (**D**) Association of histological growth pattern in incident cancers with time since last colonoscopy (Kruskal–Wallis test, *p* < 0.05 indicated by an asterisk). Green dots mark the tumors preceded by a colonoscopy identifying an advanced adenoma, orange dots mark the tumors preceded by a colonoscopy identifying an adenoma with LG dysplasia, black dots mark the tumors preceded by a colonoscopy without identification of pathological lesions. LG—low grade; UICC— Union for International Cancer Control.

**Figure 2 jcm-10-02458-f002:**
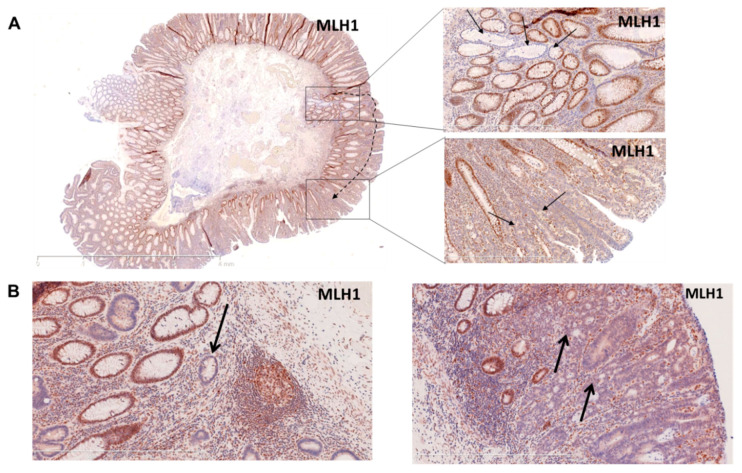
Histology images of tumor specimens with MMR-deficient crypt foci. (**A**) Resection sample with carcinoma in situ arising presumably from an MMR-deficient crypt. On the left panel, the overview of the resected sample (MLH1 staining); on the right upper panel, higher magnification of the MMR-deficient crypt (MLH1 staining); on the right lower panel, higher magnification of carcinoma in situ (MLH1 staining). (**B**) MLH1 staining revealing an MMR-deficient crypt (indicated by an arrow), on the left and another region of the same sample showing a non-invasive carcinoma in situ (indicated by arrows) on the right panel.

**Figure 3 jcm-10-02458-f003:**
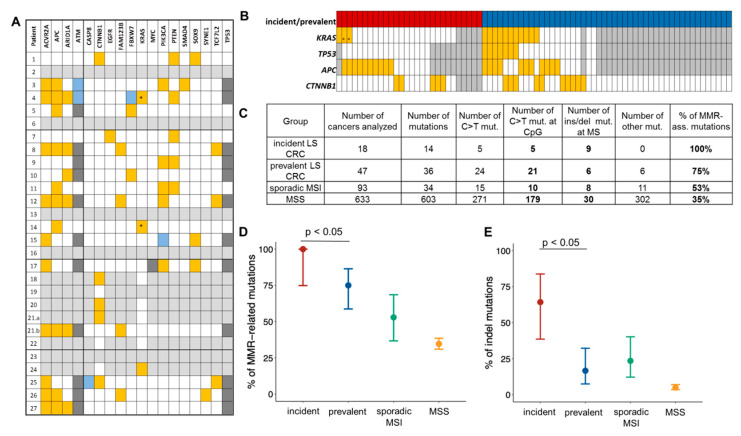
Mutational profile and MMR deficiency signatures in incident LS CRC. (**A**) Mutational characteristics of incident cancers (panel restricted to genes found mutant in at least one sample). Color code: orange—mutant, white—wild type, grey—n.a.; dark grey—known polymorphism, blue—variant of unknown significance; *—*KRAS* mutations at codons other than codon 12/13. (**B**) Mutation status of CRC genes in incident cancers analyzed in this study and prevalent cancers reported before [32,38] (for cohorts: red—incident CRC, blue—prevalent CRC, for genes—the same color code as in (**A**)). (**C**) Summary of the number of specific MMR deficiency-related mutations in incident LS CRC compared to prevalent LS CRC, sporadic MSI CRC and MSS CRC previously reported in Ahadova et al. [38]. (**D**) Comparison of the proportion of all MMR deficiency-related mutations between different CRC groups reveals a higher proportion in incident compared to prevalent tumors (100%, 95% CI: 74.85–100 vs. 75%, 95% CI: 58.7–86.4%; Fisher’s exact test, *p* = 0.0470). (**E**) Comparison of the proportion of indel mutations between different CRC groups reveals a higher proportion in incident compared to prevalent tumors (64.3%, 95% CI: 38.6–83.8 vs. 16.7%, 95% CI: 7.5–32.3%; Fisher’s exact test, *p* = 0.0068).

**Figure 4 jcm-10-02458-f004:**
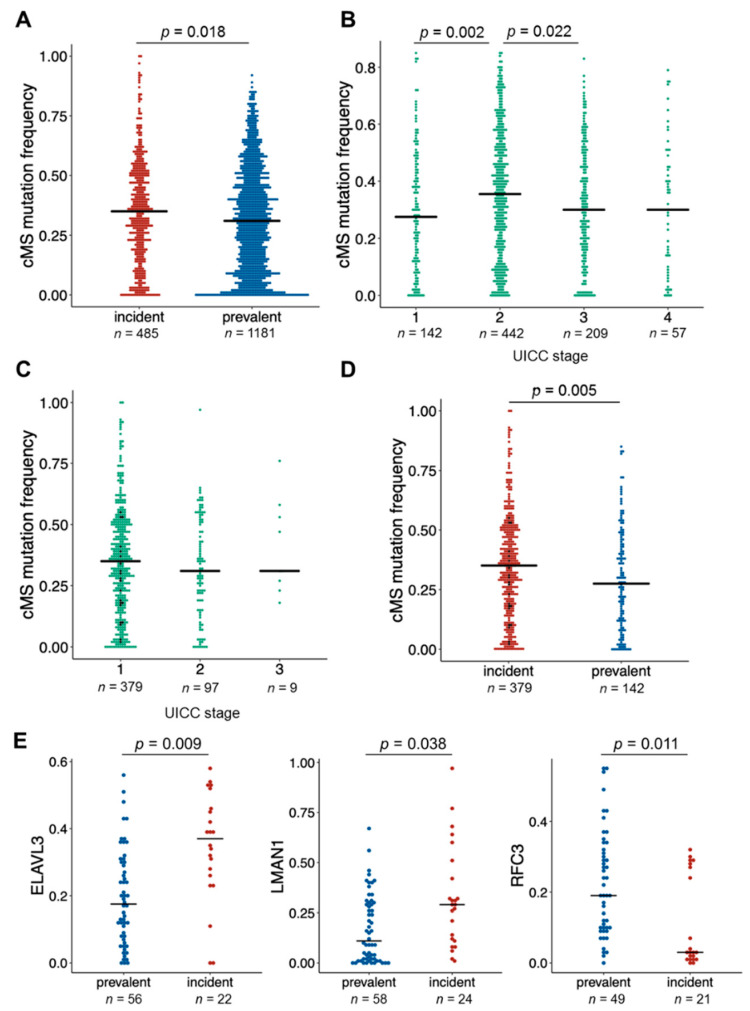
Analysis of coding microsatellite (cMS) mutations in incident and prevalent LS CRC. (**A**) cMS mutation frequency in incident and prevalent LS CRC. (**B**) cMS mutation frequency in prevalent LS CRC by UICC stage. (**C**) cMS mutation frequency in incident LS CRC by UICC stage (stage I group includes data from UICC 0 tumor, see black data points). (**D**) cMS mutation frequency in stage I incident (stage I group includes data from UICC 0 tumor, see black data points) and stage I prevalent LS CRC. (**E**) Individual cMS with significantly differing mutation rates between incident and prevalent cancers. Each dot represents one cMS locus in one tumor, and the n indicates the total number of analyzed cMS loci multiplied by the number of analyzable tumors. *p* values were calculated using a two-sample Wilcoxon test (Mann–Whitney) and corrected for multiple testing using Benjamini–Hochberg procedure.

**Figure 5 jcm-10-02458-f005:**
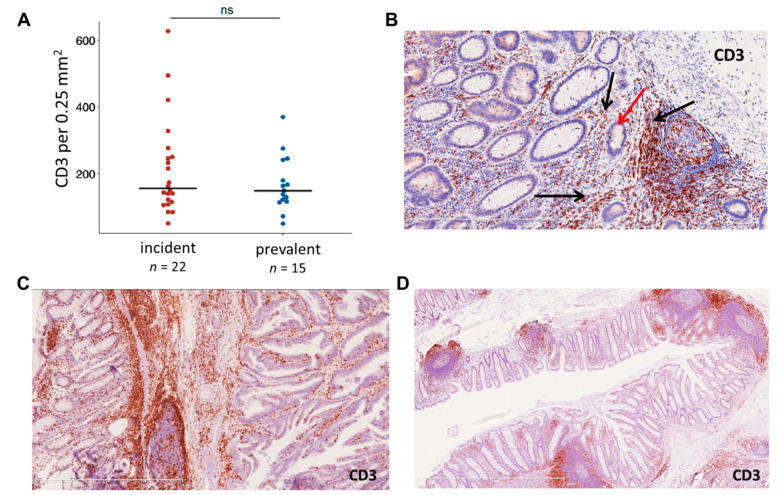
Immune infiltration with CD3-positive T cells in incident and prevalent cancers. (**A**) Immune infiltration in MLH1-associated incident and prevalent LS CRC. (**B–D**) Exemplary CD3 stainings of an MMR-deficient crypt (**B**) (see Figure 2B for the MLH1 staining, red arrow points to the MMR-deficient crypt, black arrows point at the CD3-positive T cell infiltration), tumor (**C**) and normal mucosa (**D**) regions of a transverse colon cancer specimen.

**Table 1 jcm-10-02458-t001:** Clinical characteristics of patients with incident CRC.

Patient	Age at Diagnosis	Gender	Location	TNM Stage	UICC Stage	Gene	Age at Last FU	Age at Death	Cause of Death	Months Since Last Colonoscopy	Reason of Examination	Finding at Last Colonoscopy *
1	54.6	M	splenic flexure	T2N0M0	I	MLH1	62.9			21.0	symptoms	0
2	61.7	F	ascendens	T1N0M0	I	MLH1	76.5			20.3	follow-up	advanced adenoma
3	44.2	M	descendens	Dukes A	I	MLH1	55.1			36.0	follow-up	adenoma with LG dysplasia
4	69.7	F	sigmoid	T1N0M0	I	MSH2	74.9	75.1	cardiac insufficiency	7.3	follow-up	advanced adenoma
5	63.1	F	tranverse	T3N0M0	II	MSH2	72.6	72.6	pancreatic cancer	25.0	follow-up	0
6	70.5	M	sigmoid	TisNxM0	0	MLH1	78.4			24.5	follow-up	0
7	35.5	M	caecum	T3N1M0	III	MLH1	44.3	44.3	gastric cancer	28.0	symptoms	0
8	57.3	F	sigmoid	T3N0M0	II	MLH1	63.3			30.0	follow-up	0
9	54.5	F	caecum	T2N0M0	I	MLH1	63.4	63.4	CRC	31.2	follow-up	0
10	71.6	F	rectum	T2N0M0	I	MLH1	75.1	75.1	biliary tract cancer	24.0	follow-up	0
11	41.7	F	descendens	T3N2M0	III	MLH1	44.7	44.7	CRC	23.0	symptoms	0
12	43.6	F	ascendens	T1N0M0	I	MLH1	57.5	57.5	breast cancer	26.0	follow-up	0
13	41.7	F	tranverse	T1N0M0	I	MLH1	50.1			26.0	follow-up	adenoma with LG dysplasia
14	42.4	F	tranverse	T3N0M0	II	MLH1	48.8	48.8	pancreatic cancer	24.4	follow-up	0
15	71.5	M	ascendens	T2N0M0	I	MSH2	84.1			36.0	follow-up	adenoma with LG dysplasia
16	43.6	M	tranverse	T3N0M0	II	MLH1	53.9	53.9	another CRC	26.0	follow-up	0
17	71.9	F	caecum	T2N0M0	I	MLH1	81.4	81.4	pneumonia	29.0	follow-up	0
18	69.0	F	caecum	T2N0M0	I	MSH2	69.0	69.0	postoperative complication	37.7	symptoms	0
19	42.0	F	caecum	T1N0M0	I	MLH1	58.2			39.5	follow-up	0
20	35.1	M	caecum	Dukes B	II	MLH1	64.4			37.6	follow-up	0
21.a	54.2	F	ascendens	T2N0M0	I	MLH1	63.3			30.1	follow-up	0
21.b	56.8	F	sigmoid	T3N0M0	II	MLH1	63.3			28.8	follow-up	0
22	54.2	M	ascendens	T3N0M0	II	MLH1	63.4	65.0	CUP (brain. lung)	18.0	follow-up	adenoma with LG dysplasia
23	53.8	M	ascendens	T2N0M0	I	MLH1	58.8			16.4	follow-up	adenoma with LG dysplasia
24	82.8	M	descendens	T2N0M0	I	MLH1	85.1			24.6	follow-up	0
25	55.0	F	caecum	T1N0M0	I	MLH1	66.4			39.1	follow-up	0
26	43.5	M	caecum	T2N0M0	I	MLH1	52.5			36.2	follow-up	0
27	27.2	M	caecum	T2N0M0	I	MLH1	42.9			37.9	follow-up	0

CRC—colorectal cancers, M—male, F—female, LG—low grade, CUP—carcinoma with unknown primary, UICC—Union for International Cancer Control. *—Finding at last colonoscopy column refers to the previous colonoscopy before cancer diagnosis. Advanced adenoma was defined as adenoma > 1 cm and/or with villous features and/or with high-grade dysplasia.

## Data Availability

All data presented in the manuscript are available at the Department of Applied Tumor Biology, University Hospital Heidelberg and can be shared upon request.

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
