# Peer review of "Distinct Mutational Profile of Lynch Syndrome Colorectal Cancers Diagnosed under Regular Colonoscopy Surveillance"

_jcm, 2021, doi:10.3390/jcm10112458_

Round 1
Reviewer 1 Report
The study by Ahadova et al provides important insights into the molecular etiology of incident CRCs in people with Lynch syndrome.
1) The authors should comment in the discussion on their previous work showing CTNNB1 mutations as source of an alternate tumourigenesis pathway in Lynch syndrome CRC as in this study they found very few in the incident CRC group which was not different to the prevalent CRC group. How do these new findings fit with their hypothesis of an alternate pathway of tumourigenesis in Lynch CRC that would account for incident CRCs in Lynch?
>
> 2) Also, the findings showing MMR-deficient crypts in surrounding normal of incident CRCs is not different from finding them in normal from prevalent CRCs so the comment in discussion "histologically normal MMR-deficient crypt foci were detected in the direct vicinity of two incident CRCs, providing first evidence that MMR-deficient crypts can give rise to incident CRC development in LS." Is incorrect. The results do not show that the incident CRCs have come directly from a MMR-deficient crypt so this comment should be amended.
Q. Were there any CRCs in these carriers prior to the incident CRCs?
figure 1c - add percentages for 3 slices of the pie chart
Reviewer 2 Report
In this investigation, Ahadova and colleagues focus on the study of the differences between incident and prevalent colorectal cancers in Lynch syndrome with the hypothesis that they may represent two different carcinogenic pathways. They analyzed a group of incident cancer (n=28) and prevalent cancer from a previous study (n=67), comparing the histological, immunological and molecular features. Molecular studies include a somatic panel including 30 genes, MSI status, and cMS mutational analysis previously described.
The hypothesis of the study is sound and is aimed at better understanding the carcinogenesis in Lynch syndrome. The sequence of molecular events in Lynch syndrome colorectal neoplasia remains poorly understood (it is unclear whether MMR deficiency is an early and starting event, or it appears after APC inactivation), and this study tries to shed light on this topic. However, this Reviewer thinks that there are methodological flaws that preclude to make the conclusions stated in the manuscript.
Major points:
- The main limitation of the study is that the authors are comparing two group of tumors that are essentially different in terms of tumor stage. Not surprisingly incident tumors are at an early stage compared to prevalent tumors, and this fundamental difference surely bias all the results. The more advanced stage could account for the molecular differences seen in the study. In order to control for this factor, a matched group of tumors by stage (incident and prevalent) would be desirable, but this is unlikely since most incident tumors are stage I. Moreover, mixing retrospective from different cohorts make uncontrolled bias very likely.
- A more detailed description of stage I tumors is needed. Since most tumors were stage I, how many of those were found as a polyps vs a tumor mass? What was the size of the tumors? How were these early cases treated (surgery vs endoscopy)? An example of the importance of this is seen in Figure 1A, in which what seems as a sessile polyp with a T1 is shown.
- Regarding the tumor growth pattern, it seems that only 13% of cases were described as “undermining growth pattern” while 55% were sessile and 27% pedunculated. This morphology is unclear (undermining growth pattern), and seems to refer to a non-polypoid lesion. Paris classification for polyps include polypoid and non-polypoid lesions. It is unclear what the authors are referring to. Moreover, it is surprising that despite the authors hypothesized that these incident cancers are originated from “difficult to see” lesions, most of them are sessile or pedunculated (which are easier to detect in colonoscopy).
- Adenomas are classified as low-grade dysplasia and high-grade dysplasia. Authors should not use “low-grade adenomas”. Moreover, the definition of “advanced lesion” is not described. In the GI literature, advanced lesion refers to an advanced adenoma (>1cm, and/or >10 mm, and or villous features) and/or colorectal cancer. These concepts should be clarified.
- Although the authors state that all colonoscopies previous to CRC were high-quality, the mean time before the previous exam was 27 months, which is beyond the time interval recommend by most guidelines. I wonder what would have happened if a shorter interval was used, considering that most lesions were sessile/pedunculated. Moreover, since cohort comes from a single center, I assume that timespan of the recruitment of these lesions must be wide. It is well known that colonoscopy quality in the pre-HD era was much worse than current equipment. Accordingly, the assumption that the previous colonoscopy quality was excellent may be wrong and should be tempered.
- The authors state that “…..precursor lesion more difficult to detect than conventional adenomas”, such as MMR-deficient crypts. The term conventional adenomas is mistaken. Adenomas can be polypoid and non-polypoid, as stated before following the Paris classification. The so-called MMR deficient crypts could give rise to non-polypoid adenomas, which are by definition harder to find. This should be clarified.
- In Figure 2, panel A, in situ carcinoma is used. The definition of in situ carcinoma is equivalent to high-grade dysplasia, and accordingly limited to the mucosa layer. In the picture, submucosal invasion is shown. Accordingly, it does not correspond to an in situ carcinoma but and invasive carcinoma (T1). Please correct.
- Regarding the molecular data, only a fraction of the cohort is analyzed and the frequency of KRAS and TP53 mutations could be solely explained by the difference in tumor stage.
- The results of cMS are surprising and the direct comparison of stage I incident vs stage I prevalent is interesting. The authors state the cMS mutation frequency is predicted to be higher in more advanced stages (acting as a surrogate of tumor evolution). However, in prevalent tumors, despite a higher cMS mutations in stage II compared to stage I, in stage III is significantly lower (figure 4 C). The authors conclude that MMR deficiency is likely to be an early event based on this observation, but the lack of correlation in stages III and IV somehow contradicts their hypothesis. The limited number of cases may explain these conflicting results and difficult interpretation.
Minor points:
- In the results section the aurthos state: “Here, restricting the comparison to only MLH1 carriers yielded differing proportions (4/19, 21% in incident cancer vs 8/16, 50% in prevalent cancer [49]), though not reaching statistical significance (p=0.00896). It is surprising because the p value says otherwise.
- In Figure 4, the number below the incident and prevalent cases should be better explained (example: Figure 4A: incident (n=485); prevalent (n=1181). What does this number mean? Number of mutations analyzed?
Reviewer 3 Report
This is a well-written paper regarding the clinical, histological, immunological and mutational characteristics of lynch syndromes (LS) incident cancers versus LS prevalent cancers. The idea and results of this study are of interest for the field and hypothesis generating, however the manuscript can be improved by better explaning the clinical relevance and conclusions.
Comments:
1. Firstly, the conclusion of the abstract (“preventive effectiveness of colonoscopy in LS depends on the molecular subtypes of tumors”) is different than the conclusion at the end of the manuscript (”This implies that prevention by colonoscopy may shift the molecular manifestation of LS-associated CRCs towards MMR deficiency-initiated cancers highlighting the need for preventive measures targeting MMR-deficient cells directly”). More importantly, it is not completely clear how these conclusions are based on the results of your study. The study is not designed to estimate the preventive effectiveness of colonoscopy and the use of the term ‘shift’ is not correct in this situation. Your discussion is quite lengthy, while the part of “preventive measures targeting MMR-deficient cells directly” did not seem to come forth of the discussion and was not clearly explained to the reader. In this way, the clinical relevance of this article is not sufficiently explained. A rephrased conclusion, based on your results, with more emphasis and elaboration on the clinical relevance would be more appropriate.
2. In line 58-60 you conclude there is no difference in cumulative cancer incidence, up to the age of 70 years, based on the comparison of three prospective studies and three retrospective studies. However, these studies did not investigate this question by themselves. These studies encompass different patient characteristics in different countries with different surveillance intervals. Could you comment on the validity of this conclusion?
3. It would be informative to add in the methods section why only 22 cases were assessable for tumor growth pattern in the incident cancer group and 37 in the prevalent cancer group? And for mutational profile as well (e.g., only 17 incident tumors assessable for TP53 and 21 prevalent tumors)?
4. The definition of prevalent LS cases is not very clearly described, this should be added to the method section. Could you add if these were index patients of LS families or patients who had their first (late) colonoscopy because of known LS family members? What was the reason for the colonoscopy? did they have complaints for example? It is important for the reader to be acknowledged of these facts to understand and interpret the hypotheses in the discussion. In case the prevalent group had the colonoscopy because of complaints this can partly explain the differences found.
5. Minor detail: line 23 states ‘phenotypically’ different, however you also investigated immunological and mutational characteristics.
As a remark, there are many references (70..) with around 20 references to the last two authors (Seppala and Kloor). Kloor has very interesting publications in different type of journals and the citations are mostly relevant, however this is a large number of self-citations.
Round 2
Reviewer 2 Report
The authors have been overall responsive. This Reviewer still believes that the paper would be sounder if only stage I were compared (incident vs prevalent), since comparison is largely biased by tumor stage.
Some important points still need to be addressed:
- Histological growth pattern definitions are very confusing, and macroscopic morphology is mixed between the description of a polyp and a tumor mass. The authors keep using the “undermining” term. The correct term should be non-polypoid when referring to polyps. Moreover, if as the authors state, most incident cancer were found as a tumor, it is surprising that they are described as “pedunculated”. Polyps but not tumors are usually classified as pedunculated (invasive tumors are usually a vegetative mass that rarely have a stalk). Even in prevalent cancers, the authors state that 40% of them were pedunculated. This needs clarification and reclassification following a standard classification.
- In the incident CRC authors refer to #6 as a CRC, but is described as a pTis (stage 0). In the response letter, the authors adequately explain that the tumor is a pTis. Accordingly, this case should be removed from the study because is not an invasive cancer.
- The authors have corrected the definition of advanced adenoma in Table 1, but still lacks the presence of high-grade dysplasia.
- Figure 4. CMS is used instead of cMS in the legend.
Reviewer 3 Report
Thank you for your elaborate answers and adjustments in the manuscript. The clinical relevance could possibly be further emphasized, however the idea and results of this study are of interest for the field and hypothesis generating. One final point, the conclusion in the manuscript is improved and more clear, however the conclusion of the abstract has not been changed. Could the conclusion of the abstract be aligned with the main conclusion of the manuscript?
